# OpenReview forum: "Teaching Models to Express Their Uncertainty in Words"
_TMLR — Accepted by TMLR_

### Review · Reviewer_mS55 · 2022-07-07

**Summary Of Contributions:**

The paper focuses on the area of language model calibration: Getting language models to express calibrated uncertainty about the claims they make so that users know how much they can trust the claims. E.g., a calibrated language model might say “Answer: 17. Confidence: medium high” to indicate that it is reasonably but not entirely confident in its answer.

Within this area, the contributions that the paper makes are the following:
1. **Argument for having language models express their uncertainty in words:** Previous work extracted uncertainty from the model’s logits, but the authors point out that the logit approach gives uncertainty over *token sequences*, whereas what the authors want is uncertainty over *claims* (and a claim might be expressible with many token sequences)
2. **An approach for getting models to express their uncertainty in words:** The authors use supervised fine-tuning. To get the labels that provide the supervision, the authors use the model’s accuracy on the category that each specific example belongs to. E.g., if the example is an addition problem and GPT-3 gets 70% accuracy on such problems, the supervised confidence label might be “70%” or “high.”
3. **A dataset called CalibratedMath for testing how well models can express verbalized probability:** This dataset contains a range of math problems for which a model must provide both an answer and a confidence level. Because GPT-3 has varying levels of performance on different types of math problems, this dataset supports diversity in the label distribution that models should produce for expressing their confidence. In addition, the dataset is set up in a way that gives a distribution shift from fine-tuning to evaluation, so that models cannot succeed just by memorizing the classes of problems they have seen.
4. **Experiments showing that GPT-3 can verbalize its uncertainty reasonably well:** The experiments show reasonably strong in-distribution and out-of-distribution uncertainty levels; the out-of-distribution performance suggests that GPT-3 is indeed expressing uncertainty, not just memorizing the categories it has seen. In some cases, verbalized probability outperforms other methods and baselines; in other cases it does not.
5. **Experiments further probing GPT-3’s verbalized probability abilities:** The authors show that GPT-3’s success at verbalized probability cannot be explained just by converting logits to words or by using several shallow heuristics. They then show that the success may be explained by pre-existing latent representations of uncertainty in GPT-3’s representations (i.e., information that is present in GPT-3 even before fine-tuning).


**Broader Impact Concerns:**

See “Requested Changes 1.2.” I don’t think that a Broader Impact Statement is necessary, but it could be added as a way to address some of the concerns I had.

**Requested Changes:**

As it currently stands, I would recommend rejecting the paper due to the weaknesses listed above. If those weaknesses were addressed, I would then be much more likely to recommend accepting the paper (though I can’t guarantee it, because it would depend on how convincing the responses are). Because the problems I listed above are fairly fundamental ones, I would not expect the paper to solve all of those problems, but I would expect it to at least acknowledge them. Specifically, here are the changes that I think would be reasonable and that would go a long way toward addressing those concerns:
1. To address Weakness 1 and Weakness 2: The paper could add in some discussion of the fact that we may never be able to fully trust language models alone, without verification from trusted external sources.
    1. Some points that could be made along these lines include the following (I’m not saying you should make all these points - I’m just listing ideas that occur to me in case you find them useful):
        1. The paper could mention scenarios where models might be confident but wrong
        2. The paper could suggest that any usage of confidence scores could be accompanied by a caveat that a high confidence does not necessarily mean it is true.
        3. The confidence scores could be given a ceiling of, say, 90%, so that there would always be some uncertainty reflected.
        4. Confidence scores could be accompanied by reference to external databases; e.g., you could get a confidence score of 100% if and only if it can be verified in a trusted external knowledge base.
        5. Future work could investigate the specific cases I listed like misconceptions or outdated information; it might be possible to teach models to recognize these types of scenarios and reduce their confidence accordingly. (E.g., you could teach models that information about who is the leader of a country is liable to becoming outdated). To be clear, I’m not at all saying that you should run these experiments - just something that you could mention as potential future work.
    2. Some or all of these points could be made in a Broader Impact statement. Specifically, the statement could say something like: “A potential concern with providing confidence scores is that sometimes a model might assign a high confidence to false statements, which would then mislead users. Here are some ways that this concern could be addressed: [X,Y,Z].” On a more positive note, the Broader Impact statement could also point out how confidence scores will help users recognize weaknesses of current models, so they have the potential to reduce the harms of such models (as long as we remain open about the fact that the confidence scores themselves might be flawed).
2. To address Weakness 3: The paper could include a bit more discussion of this concern, beyond the oblique mention of it in Section 3.1. Even better would be brief mentions of ways that future work could build toward evaluating confidence scores at the level of individual examples rather than just groups of examples (but I don’t have any suggestions about how this would be done).

All of the remaining points listed below are not critical for securing my recommendation - they would simply strengthen the work in my view:
1. Figure 5: Does “training” refer literally to training examples that models have seen, or to withheld examples that are in-distribution?
2. Section 3.3: It took me a couple re-reads to figure out the point of the stochastic few-shot experiment. The motivation that is given is “in order to learn more about how verbalized probability generalizes”, but as I understand it the motivation is more like “in order to understand the extent to which verbalized probability is already present in the pre-trained model vs. learned via fine-tuning”? If this interpretation is correct, it might be worth it to clarify.
3. On page 9: It says “cannot be fully explained by these heuristics”: I would suggest changing to something like “cannot be fully explained by these heuristics, though the possibility remains that it could be explained by other heuristics that we have not considered.”
4. On page 9: You used a GPT-3 model “finetuned for semantic similarity.” Why did you use this version instead of a non-finetuned one? Is it the case that the GPT-3 API does not allow you to get embeddings from the non-finetuned one?
5. Since Section 4 has only one subsection, you could get rid of the subsection 4.1 heading and just call Section 4 “Discussion: Directions for future work”


**Strengths And Weaknesses:**

To summarize my overall view: I think that the premise of the paper is very well executed, with a strong set of interesting experiments; but I am skeptical about some aspects of this premise, making me skeptical about the usefulness or informativeness of the paper overall.

In more detail, the strengths are:
1. The paper is motivated by a very important problem in current NLP, namely hallucinations by language models.
2. The paper presents a clear argument for why expressing uncertainty in words is better than previous approaches for addressing this problem
3. The paper introduces a well-motivated and thoughtfully-constructed dataset for evaluating whether models can express their uncertainty in words
4. Using this dataset, the paper provides experiments showing that GPT-3 can perform reasonably well at verbal probability
5. (Perhaps the biggest strength, in my opinion): The authors do not just take their initial experiments at face value but instead conduct several careful analyses testing whether their results can be explained by factors other than uncertainty, such as shallow heuristics.
6. The paper is overall extremely clearly written. This specific area is not one I am familiar with, yet I found the paper very easy to follow.

Before I state the weaknesses, let me be up front about some of my own views: I do not believe that language models can ever be trusted as sources of factual statements because language models are fundamentally learning about linguistic form, not learning about meaning (below I elaborate more on why I don’t think that confidence can address this concern). I don’t say this to be antagonistic - I just wanted to be up front about how my views may have affected my assessment of the paper, so that the editors can factor this in when considering my review.

With that said, here are the paper’s weaknesses, in my view:
1. The “confident but wrong” concern: I am skeptical that the problem of hallucinations can be solved by language models themselves (as opposed to checking the model’s statements against trusted external databases). The reason is that language models could be confident but wrong; thus, the fact that a model is confident is not enough to conclude that we can trust it.
    1. Here are several specific cases where language models would be likely to be confident but wrong:
        1. Common misconceptions that are presented as fact in the training data: e.g., models might be confident that “Marie Antoinette is the source of the quote ‘Let them eat cake!’”
        2. Outdated information: e.g., models trained before 2022 would probably be confident that “Angela Merkel is the Chancellor of Germany,” even though this statement is now false
        3. Fiction: models’ training data usually does not clearly differentiate between factual and fictional text, so models might confidently believe statements that are made within a fictional context, such as “Certain people are born with superpowers such as x-ray vision”
        4. Conspiracy theories and deliberate misinformation: Certain false statements might be widely presented as truth in training data derived from the internet (e.g., “The Earth is flat”)
        5. Social biases: Language models have been shown to have many social biases (e.g., https://arxiv.org/pdf/2009.11462.pdf?fileGuid=zX3zH5DBtXQa0ktq). Thus, models might be highly confident in false, prejudiced statements like “Women cannot be good computer scientists.”
    2. I am skeptical that models would be able to differentiate from cases where they are confident and right vs. cases where they are confident and wrong. For this reason, I think it is unlikely that the hallucination problem can be solved using language models themselves. And the CalibratedMath dataset does not address this concern, because math is one domain that is unlikely to be affected by the 5 situations listed above.
2. Model confidence vs. user confidence: This concern is closely related to Weakness 1: The paper does not clearly differentiate two types of confidence: (i) The model’s confidence in its answer, and (ii) how confident users can be in the model’s answer. I will refer to (i) as “model confidence” and (ii) as “user confidence.”
    1. I believe that much of the wording in the paper suggests *model confidence*. For example, the title is about teaching models to express “their” uncertainty; page 1 says “models should be calibrated about _their own_ uncertainty”; and Figure 1’s caption says “GPT-3 is scored on the calibration of its confidence.”
    2. However, I believe that the goal of the paper is really *user confidence*: the motivation is about enabling users to know whether they can trust a statement made by a model, and the confidence is evaluated with respect to the ground truth.
    3. As discussed under Weakness 1, I find it plausible that models can express model confidence, but I find it unlikely that they can express user confidence. This is a problem if the goal is user confidence (which I believe is indeed the goal). To the extent that model confidence and user confidence are aligned, the problem is solved; and the results of this paper suggest that, for the CalibratedMath domain, model confidence and user confidence can be aligned. However, as discussed under Weakness 1.1, there are several cases where model confidence may not be aligned with user confidence, and the CalibratedMath dataset is unlikely to address these cases, meaning that the paper as it stands does not address one of my main concerns with the general direction of using a language model to verbalize user confidence.
3. Confidence in categories vs. individual examples: With the way that confidence is presented here, confidence can be reliably interpreted for groups of examples, but not for individual examples. This weakens the usefulness of this type of confidence, given that the motivation on page 1 is to enable users to know “how much to trust a given statement.”
    1. As an example, consider the following scenario:
        1. Suppose that there are 8 examples (A,B,C,D,W,X,Y,Z) where the model gets the first 4 correct and the second 4 incorrect:
            - Correct: A,B,C,D
            - Incorrect: W,X,Y,Z
        2. Now consider these 2 different ways of assigning confidence scores:
            - Assignment 1:
                - 100% confidence: A,B
                - 50% confidence: C,D,W,X
                - 0% confidence: Y,Z
            - Assignment 2:
                - 100% confidence: C,D
                - 50% confidence: A,B,Y,Z
                - 0% confidence: W,X
        3. Assignment 1 and Assignment 2 both get identical scores under MSE and MAD. Yet they assign very different confidences to individual examples (e.g., example A getting 100% confidence vs. 50% confidence).
    2. As the authors discuss in Section 3.1, their setup is designed in a way that makes models likely to assign a uniform confidence within categories of examples, rather than being sensitive to individual variation within a category: “We recognize that this approach can lead to suboptimal labels. For example, it might use a low-confidence label for “10 × 10 = 100” because most two-digit multiplications are hard for GPT-3.“ The authors note that this approach nonetheless “works well enough for our purposes”; but this statement is based on holistic evaluations over groups of examples. Thus, I believe the concern remains that, for the purpose of assessing confidence in particular individual examples, the approaches presented in this paper might not be informative.

---

> ### Author Response · Authors · 2022-07-27
> **Response to the claim that models cannot possibly be well calibrated (“confident but wrong”)**
>
> (We thank Reviewer mS55 for their detailed review and engagement with our paper.)
>
> Reviewer mS55 is skeptical that the problem of hallucination in language models can ever be solved by language models themselves. They write:
>
> “I am skeptical that models would be able to differentiate cases from where they are confident and right vs. cases where they are confident and wrong”.
>
> First, whether models can learn to be better calibrated is an empirical question. It’s important to investigate this with experiments. In many domains (e.g. programming and high-school math), language models have performed better than people expected. So we disagree that this counts as a “weakness” of our paper.
>
> Second, it seems Reviewer mS55’s main point is that language models cannot reliably answer certain questions (e.g. "Who is the current prime minister of Israel?” or “Did it rain today in New York?") because they have outdated information. But not having access to up-to-date information does not prevent the model being *calibrated* (more on this below).
>
> Third, we will explain why it’s plausible that models can learn to differentiate “confident and right” from “confident and wrong”. Suppose a model assigns confidence 95% to the false answer A=“The Earth is flat” after a prompt P, and is also overconfident about other answers that appear in fringe theories or conspiracies. If we finetune the model to improve calibration, the model should learn to avoid assigning high probability to these conspiracy-theory answers. The model can learn this because the prompt together with the conspiracy-theory answers include features that the model can recognize (e.g. phrases and ideas that are more common on conspiracy-theory forums than on Wikipedia or BBC).
>
> Likewise, we expect models can learn to reduce their confidence in common misconceptions like “Humans use only 10% of their brain”, which are common on social media but don’t appear in any scientific papers, Wikipedia, or high-quality journalism.
>
> Models could also learn to reduce their confidence in time-sensitive questions (e.g. “Did it rain in London today?”). We agree with Reviewer mS55 that models cannot give reliable answers to all time-sensitive questions without the ability to query external databases. But this doesn’t prevent models being well calibrated about such questions. After fine-tuning, the model could learn to distinguish questions with quickly changing answers (“Did it rain in London today?” or “Who is the PM of Israel?) and those with slow changing answers (“What are the two largest political parties in the U.S?”).
>
> Some additional points:
> 1. We don’t expect that a model like GPT-3 can be fine-tuned to obtain *perfect* calibration in all the domains mentioned by Reviewer mS55. But it’s plausible that calibration can be substantially improved over the pre-trained model.
>
> 2. Recent results have shown that calibration on diverse multiple-choice questions improves with model scale: https://arxiv.org/abs/2206.04615, https://arxiv.org/abs/2207.05221.

---

> > ### Comment · Reviewer_mS55 · 2022-08-06
> > **Confident but wrong**
> >
> > I completely agree that all of these things are empirical questions, and the claims that you make are all plausible stories for how calibration could overcome these issues. Nonetheless, it still strikes me that one of the core challenges for calibration will be to deal with these scenarios, so the fact that this paper does not engage with these situations is still in my view a shortcoming.
> >
> > To expand on this concern:
> > 1. We want for models to be well-calibrated.
> > 2. There are some scenarios where it is easy to be well-calibrated, and others where it is hard.
> > 3. As a rough first pass, I would guess that calibration is easy when the following heuristic holds, but hard when the heuristic fails: “Assign confidence to a statement in proportion to how clearly that statement was made in the pretraining data.” (So high confidence if it appeared exactly, medium confidence if something similar appeared, etc.)
> > 4. The test cases in this paper are all of the sort where the heuristic works; the paper does not include cases where that heuristic fails (cases like misconceptions, outdated info, etc.)
> > 5. This raises the concern that it might not actually be possible to fine-tune models to express well-calibrated uncertainty in words. Perhaps it only appears possible to do this because the model’s calibration was only tested on cases where calibration is relatively easy. Perhaps what models are expressing in words is not actually calibration but rather “degree to which the statement was made in the training set.”
> >
> > I agree that it’s plausible that the harder cases *could* be addressed, but this paper does not attempt to address them, and without an actual test it’s hard to be certain either way.

---

> > > ### Author Response · Authors · 2022-08-13
> > > **Disagree that this is a weakness**
> > >
> > > The reviewer thinks there might be a challenge for *ever* creating well-calibrated LMs because of the “confident but wrong” problem. They give no empirical evidence that this is an insurmountable problem. We think there is already some contrary empirical evidence: the results in this paper on Big-Bench and TruthfulQA (https://arxiv.org/abs/2207.05221) that show calibration improving with model size on some of the kinds of question the reviewer thinks are problematic.
> > >
> > > Moreover, we think that calibration being (potentially) difficult or impossible for some types of question does not mean that any paper on calibration must address that type of question. We conducted a thorough investigation of a particular domain (simple math problems). We fully acknowledge that this domain is not representative of all other QA domains. However, it has many advantages as a toy domain. It’s easy to understand; the questions can be auto-generated; GPT-3 differs greatly from humans in which question types it does poorly on; and the categories range in difficulty from 0% to 100% accuracy.
> > >
> > > We are excited to see followup work that explores domains outside of math. We think math is an appropriate domain for a first paper in this area.

---

> ### Author Response · Authors · 2022-07-27
> **Response to User/Model Confidence**
>
> We disagree with the Reviewer that this constitutes a weakness. Here is our response:
>
> 1. Our goal is to fine-tune a model to provide well calibrated confidences. If a model does this, then humans will know when to trust a model. For example, if a calibrated model assigns low probability to its own answer to a question, then humans would know to not trust the answer.
>
> 2. The reviewer says, “the goal of the paper is really user confidence [as opposed to model confidence]”, but we find this unhelpful and misleading. As we’ve already noted, the confidence of a calibrated model is a guide to how much the user can trust a model. So it’s correct for us to focus on the model’s confidence.
>
> 3. The reviewer's point (3) is based on the “confident but wrong” concern, which we already discussed in a different comment.

---

> > ### Comment · Reviewer_mS55 · 2022-08-06
> > **User/Model Confidence**
> >
> > Thank you for the response! Let me clarify the 2 potential interpretations of the paper's goals that I had in mind:
> >
> > - Interpretation 1: Pretrained GPT-3 has some pre-existing confidence in its statements (this is what I meant by “model confidence”). The goal of this work is to make that confidence explicit by getting GPT-3 to verbalize it. So, under this framing, the fine-tuning is not affecting GPT-3’s confidence but rather just teaching it *how to express* the confidence that it already has.
> >
> > - Interpretation 2: The goal is to get GPT-3 to accurately inform users about how much they can trust a given statement from GPT-3 (this is what I meant by “user confidence”). So, under this framing, the fine-tuning is not just teaching GPT-3 how to express its confidence, but also teaching GPT-3 what its confidence should be. Maybe another way to phrase this, in line with your point (2), is that the goal is to bring the model’s confidence in line with user confidence.
> >
> > It took me a couple re-reads of the intro to figure out that Interpretation 2 is the intended reading of the paper; my initial interpretation was Interpretation 1, because I think that some of the wording suggests this (particularly the title being “Teaching Models to Express *Their* Uncertainty in Words”; I actually think a lot of my confusion would have been avoided if the word “their” were not in the title, or if it were changed to “calibrated,” since to me “their” suggests that the uncertainty in question is something pre-existing that we got from the model, not something new that we are teaching to the model). And so that's why I said that the paper's goal is not about model confidence, since I intended for "model confidence" to refer to something pre-existing.
> >
> > Thank you for clearing this up for me! I do think that the paper could benefit from having this point clarified, just so other readers don't get confused like I did.

---

> ### Author Response · Authors · 2022-07-27
> **Response to "Confidence in categories vs individual examples"**
>
> The reviewer writes that “for the purpose of assessing confidence in particular individual examples, the approaches presented in this paper might not be informative”.
>
> We disagree with this. In our paper, finetuning improves calibration on the two evaluation sets (both MSE and MAD metrics). This means that the model’s confidences for individual examples become more informative about whether the answers are true or false.
>
> In general, calibrated models will assign similar confidences to similar questions/answers. For example, if a model was totally ignorant about quantum mechanics, then it should assign 50% probability to all true/false exam questions about quantum mechanics. (The only way to avoid assigning similar confidence to similar questions/answers is if a model says 100% confidence for every true answer and 0% for every false answer).
>
> In the paper, we finetune the model using fixed confidence labels for each of ~100 categories of question (which vary based on the math concept involved, # of digits, and formatting). We point out that this *may* lead to models producing less informative (in terms of MSE) confidences than they are capable of. However, such confidences would still be informative about individual examples (contrary to what the reviewer says). Moreover, confidences would only be less informative in-distribution, whereas our evaluation is off-distribution. Concretely, there are ~100 categories of question on the addition/subtraction training set. The questions on evaluation sets all involve different mathematical concepts (e.g. prime numbers) and may allow for multiple correct answers (“Name a prime number below 100”). Thus, the model cannot “know” which category questions in the evaluation sets will fall under because it lacks labeled examples (and the categories cannot be easily deduced from background knowledge).

---

> > ### Comment · Reviewer_mS55 · 2022-08-06
> > **Categories vs individual examples**
> >
> > I agree completely that increased MSE or MAD shows that the model is better calibrated overall, which also entails that its predictions become more informative about individual examples in general. However, it does not mean that the model is doing a better job at capturing meaningful differences within categories (at least, I don’t think so - please correct me if I’m wrong!), and this is what I'm focusing on.
> >
> > E.g., suppose we give a model a true/false quiz about national capitals. And suppose the model in general is bad at knowing national capitals; but its training data included the website of the Kenyan government stating “Nairobi is the capital city of Kenya,” so it should know this fact with high confidence due to the authoritative source. If we consider the following 2 models, in principle it seems like Model A should score more highly, but I believe that the metrics considered here would give the same scores to both:
> >
> > - Model A: Assigns 100% confidence to the Nairobi question (which the model knows, and gets right because it knows it) and 50% to all others (which the model is guessing randomly on)
> > - Model B: Assigns 100% confidence to the Paris question (which it gets right, but only because of good luck) and 50% to all others (including the Nairobi question, which it gets right because it actually “knows” it).
> >
> > Maybe a different way to put this point: A higher MSE or MAD shows that in general scores are becoming more informative for individual examples, but it doesn't necessarily mean that the model is getting better at capturing differences within a category; it might instead mean, e.g., that the model is doing a better job of assigning the right confidence to a category, but still using the exact same confidence uniformly across all category members.
> >
> > I don’t think I follow the argument about how out-of-distribution evaluation should help on this particular issue. I completely agree that the model can’t just be learning a set of categories with corresponding confidence labels (since the OOD case has new categories), but it could still be learning “when I see an example, I should abstract away from this specific example and instead make up a higher-level category that this example belongs to, and then produce my confidence for that category.” This strategy would mask differences within categories. In other words, the training scheme (which uses the same score for all members of a category) could discourage a model from considering the factors that would otherwise enable it to capture within-category variation (like the authoritativeness of the model’s source for a given fact).

---

> > > ### Author Response · Authors · 2022-08-13
> > > **We disagree this is a weakness**
> > >
> > > Regarding the Model A and B example:
> > > This might be a problem if our test set was very small. But we evaluate calibration on 10k questions (which vary considerably in difficulty).
> > >
> > >
> > > "when I see an example, I should abstract away from this specific example and instead make up a higher-level category that this example belongs to, and then produce my confidence for that category."
> > >
> > > There is no evidence that our models are doing this. And it’s not clear to us why it would be a problem if they did. (What matters is how well calibrated the models are.)
> > >
> > > Also, the calibration performance of the 50-shot model is close to the finetuned model. The 50-shot model sees so few examples that it cannot learn that “examples in the same category should get the same probability”.

---

### Review · Reviewer_st8A · 2022-07-12

**Summary Of Contributions:**

This is a nice paper on an important topic. I would be happy to see it appear in TMLR, with revisions.

The indirect logit and verbalized uncertainty are clever ideas. I am glad the authors chose to use OOD evaluations. I am also pleased they tested heuristic accounts. The results are promising, although there's room for improvement in future work.

I do not closely follow the work on calibration, so I am not in a position to judge the novelty of the work.



**Requested Changes:**

The methods developed in the paper perform better than baselines, but it's hard to judge how good the model calibrations are in any objective sense. As the authors say, the MSE loss will not be zero, even for a perfectly calibrated model. It's also not straightforward to interpret the MAD estimate. For MSE, the authors could report an upper bound for an oracle model. For MAD, it would be useful to see the oracle level for having the category right but not the exact numbers, or some way to interpret absolute performance.

It's odd the paper ends with a related work section rather than a conclusion. This makes related work seems like an after thought.

Minor comments
Figure 4: : legend is covering baseline
Figure 12: caption title doesn't apply to right panel

**Strengths And Weaknesses:**

Strengths
- important topic
- interesting method
- well-written

Weaknesses
- difficulty in judging objective performance

---

> ### Author Response · Authors · 2022-07-27
> **Baselines**
>
> We thank the reviewer for their helpful comments and suggestions.
>
> We agree with the reviewer about the difficulty of interpreting the calibration results. These are very helpful suggestions and we will consider adding them to the paper. (Another baseline we considered was "human performance" but this seemed to tricky to do for simple math questions.)
>
> Regarding "Related Work", we tried to cover some of the most relevant work in earlier sections. (There are 16 citations *before* the Related Work section).

---

> > ### Comment · Reviewer_st8A · 2022-08-22
> > **Requested changes**
> >
> > I would like to see these changes before I can recommend the paper for acceptance.

---

### Review · Reviewer_oCZ4 · 2022-07-17

**Summary Of Contributions:**

This paper proposes to finetune a language model to output *verbal* estimates of its own uncertainty to math questions.

As the authors note, traditional approaches to model calibration and uncertainty estimation typically operate on model logits, but this is fundamentally flawed: if there are many plausible answers to a question, the logits on individual answers might be low, even if the model is broadly confident in the set of correct answers.

Verbalized uncertainty proposes to overcome this: if we can construct a manual dataset of (question, answer, groundtruth certainty) labels, we can finetune GPT-3 itself to predict its uncertainty, presumably making use of the semantics of the questions and answers as well as its own representations which might encode epistemic uncertainty.

To obtain this calibration dataset, the authors ask for GPT-3's predictions on a set of addition/subtraction problems, composed of *subtasks* of varying difficulty (defined by the # of digits, format of the numbers, etc). They estimate the calibration probability for a specific subtask by averaging performance across 100 samples from that subtask. These are binned into 5 categories: "lowest", "low", "mediujm", "high", "highest", which are then used as finetuning targets for GPT-3, though note that these actual labels are not used specifically, rather nonce words are used ("john", "sam", "matt", "dan", "tom") which are then mapped back to the original English labels. This apparently results in better performance.

After fine-tuning on this dataset, they then evaluate uncertainty estimates under domain shift: on a dataset of multiply/divide questions, for example, or a Multi-Answer dataset where each question has multiple correct answers (e.g. "name any prime number under 50"). The authors show that the verbalized probability estimates generalize reasonably well under some domain shifts (e.g. from add/subtract to multi-answer) but perhaps less well under others (e.g. from add/subtract to multiply-divide).

The authors explore a few hypotheses as to what information is being leveraged to output verbalized probabilities. Perhaps most importantly, they tackle the thorny question of whether GPT-3 is actually using
 its own epistemic uncertainty (represented in its learned representations somewhere) to produce probabilities, or simply learning to match superficial task features to the computed probabilities (e.g. more digits -> less certainty). They do this by comparing a logistic regression model trained on the subtask features (e.g. digit length) to predict the probabilities, showing that this doesn't do as well, though I think more could be done towards answering this question (see Weaknesses).

Overall, the contribution of this paper seems to be proposing to learn a calibration model given pretrained semantics and question and answer contexts, rather than just from model logits. Similar ideas exist in the literature (see Zhang et al., 2021 and Mielke et al., 2022), though it has not quite been applied in this kind of setting, with the pretrained semantics provided by large language models.

**Broader Impact Concerns:**

Again, I have a large broader impact concern of this paper's wording implying an emergent meta-cognitive behavior of large language models ("learning to express its own uncertainty in natural language") rather than a relatively simple calibration task implemented as discrete 5-class supervised learning.

**Requested Changes:**

Overall I am currently not ready to recommend the paper outright, but I look forward to the other reviews and the author response to help update my views. Perhaps the biggest issue for me is:

## (Critical) Framing
To summarize the above: in my opinion, a sensible description of this paper's contribution is to calibrate a model not by using model uncertainty (e.g answer logits) or other superficial features (e.g. Jiang et al., 2021), but by building a calibration model directly on the semantics of the problem domain (as revealed in the question and answer texts) as well as the model's internal representations. This is not a completely novel idea (Zhang et al., 2021; Mielke et al., 2022), though the lack of novelty is not a fatal weakness of the paper.

The trouble I have with this paper is that it is framed as something fundamentally different than these existing calibration studies. That the model itself (GPT-3) is moonlighting as the calibration model is a neat parlor trick, which probably helps the authors take the jump that GPT-3 is "learning to express uncertainty about its own probabilities", but I would really strongly discourage this framing, for the unnecessary hype/anthropomorphization it creates, or at the very least, more precisely evaluate whether it's truly required to have GPT-3 itself serve as the calibration model (versus some other reasonable calibration model).

I of course invite the authors and other reviewers to disagree with me if they think I have misunderstood the contributions here.

## More baselines and comparison to existing calibration methods

(Critical) At the minimum I'd like more description of the logistic regression/heuristic baseline, and even better, more stringent experiments that measure to what extent having the latent semantics actually matters for calibration estimates.

(Good to have) Comparison to other calibration methods would also strongly strengthen the paper. In particular, a key contribution this paper stands to make is to more precisely evaluate whether the extra information improves, or can be used in tandem, with more traditional probabilistic approaches to model calibration.

## Minor suggestions/questions, in decreasing order of importance
- How many possible answers are there on average for the multi-answer dataset? It seems adversarially designed for poor performance for model logits. A cynical reading of the main results are that "verbalized probability underperforms every calibration method except for the one specifically designed with many correct outputs." It would be great to have at least another data split to disentangle whether the benefits of verbalized probability extend beyond the extreme multi-answer case.
- It would be great to have the baseline of train calibrator on multi-answer -> test calibrator on addition/multiplication. There may as well be every train/test split option in the paper.
- Why not use something simple like platt scaling on the model logits? Zero-shot may be an unrealistic baseline.
- I'd prefer that Figure 2 at least mentions in a footnote somewhere that he model is not actually trained to output "Medium", but rather a random name (which is then mapped on to Medium), otherwise this figure becomes slightly misleading.
- Why alternate between EV uncertainties presented in some experiments and greedy in others? Why not keep this consistent?
- I would love to see a discussion on the similarities and differences between this work and https://arxiv.org/abs/2207.05221 (which came afterwards of course, so this new paper is not relevant to my overall recommendation of the current paper). In particular this work tries some zero-shot uncertainty estimation (as I mentioned above) as well as learns calibration estimators directly on the model's internal representations (the "I Know" classifier)
- Footnote 7 Typo - "finetund"

# Missing References

[Sabrina J. Mielke](https://arxiv.org/search/cs?searchtype=author&query=Mielke%2C+S+J), [Arthur Szlam](https://arxiv.org/search/cs?searchtype=author&query=Szlam%2C+A), [Emily Dinan](https://arxiv.org/search/cs?searchtype=author&query=Dinan%2C+E), [Y-Lan Boureau](https://arxiv.org/search/cs?searchtype=author&query=Boureau%2C+Y). Reducing conversational agents’ overconfidence through linguistic calibration. NAACL 2022.
https://arxiv.org/abs/2012.14983

[Shujian Zhang](https://arxiv.org/search/cs?searchtype=author&query=Zhang%2C+S), [Chengyue Gong](https://arxiv.org/search/cs?searchtype=author&query=Gong%2C+C), [Eunsol Choi](https://arxiv.org/search/cs?searchtype=author&query=Choi%2C+E). Knowing More About Questions Can Help: Improving Calibration in Question Answering. Findings of ACL 2021. https://arxiv.org/abs/2106.01494

**Strengths And Weaknesses:**

# Strengths
- This paper tackles the important problem of calibrating language models, with clearly relevant impacts in robustness and safety. It is clear that "some individuals in TMLR's audience [would] be interested in the findings of this paper" (acceptance criterion 2).
- The core idea that one can build a calibration model using the actual semantics of the questions and answers, and its own internal representations, to predict uncertainty, rather than model logits, is clever and viable, especially for models without access to logits (as the authors note). It is not completely novel but has not yet been applied in this large language model setting.
- I think the question of whether a model is able to "meta-learn" its own epistemic uncertainty---to make calibration predictions from its own internal states---is a fascinating and philosophical one, though one that could be explored further in this paper. Admittedly, this is a very difficult question!
- Evaluating calibration under domain shift is the right thing to do, and this paper does it.
- The CalibratedMath dataset should be a useful contribution for people looking at calibration, since it's often difficult to obtain calibration estimates for open-ended tasks.

# Weaknesses
Unfortunately I have a few issues with the paper which makes me hesitant to recommend it outright:

## Framing

In the author's words:

> This is the first time a model has been shown to express calibrated uncertainty about its own answers in natural language.

I have a lot of issues with this claim. The first is the use of the term *natural language*. The calibrator is trained on a 5-class classification task, producing labels "john", "sam", "matt", "dan", "tom", which map back to English labels of probabilities (low, medium, high)—as the authors note, using the actual semantic labels actually *hurts* performance. If this were natural language generation, then an ImageNet classifier can be said to generate natural language because it predicts classes which happen to have names. And indeed **any** calibration method in the literature, going all the way back to Platt scaling, can be said to output "natural language" if the probabilities are just binned into the 5 categories used here.

The second is the wording, especially the word "shown", which seems to imply that the authors have **discovered** the emergent behavor of uncertainty estimation in these models. This is misleading; the model was *finetuned* (read: *forced*) to express uncertainty labels (and again, not even actual semantic labels, but literally nonce labels).

Moreover, there is no actual requirement of this method that the model itself learns to predict its probabilities; it can be any auxiliary model, small to large, pretrained or not. This means that the proposed approach is actually a relatively straightforward calibration method, that uses pretrained semantics of the questions and answers to predict uncertainty scores, rather than model logits. More precisely, it shares similarities with both Zhang et al., 2021 and Mielke et al., 2022, which both propose to learn calibration models from question/answer contexts and internal representations, respecitvely (which Mielke et al. dub "metacognition"), yet this work is not mentioned in the paper.

To claim that this is a qualitatively different phenomenon from much of the calibration literature—to imply that this a case study of a model which has somehow organically "developed" the ability to express its own uncertainty—feels downright irresponsible, and will only add to the already inflated "AI hype" media cycle.

If the model was able to express zero-shot uncertainty, perhaps without even prompting, I would be more comfortable with this framing (and this would be a very interesting experiment to run). The k-shot experiments get at this idea, though it is not the main focus of the paper.

## Are you sure GPT-3 isn't using superficial statistics/is it really important to have GPT-3 calibrate itself?

The claim that this paper seems to be getting at—that GPT-3 itself is learning to express uncertainty based on not only the semantics of the questions and answers, but a certain "meta-awareness" of its own representations—is a very bold claim for an important question that needs not only the framing fixes discussed above, but also more thorough investigation than is currently presented in the paper.

Again, the core idea of this paper does not require that GPT-3 itself is the calibrator. The authors correctly propose to use a logistic regression baseline as an alternative calibrator. Yet I find this baseline quite weak and underexplained. First, more detail is required on the kinds of heuristic features and their inputs to the logistic regression baseline. Surely there is a domain shift in the heuristic features that logistic regression likely can't handle well at all, which could explain the poorer calibration estimates for the eval domains?

A stronger and more interesting set of baselines would be:

- Calibrating with non-pretrained (random) GPT-3. This aims to keep the model capacity constant (unlike logistic regression) while removing the latent question semantics. Performance of this baseline would give a much clearer sense of whether the model is using the latent question features.
- Calibrating with a different large language model (e.g. GPT-J, GPT-2, or even a different seed). This tries to get at the vaguer question of whether or not it's important to have access to the models question features itself, or rather just generic semantics about the questions and answers (e.g. that multiplication is generally harder than addition, for example). If alternative models serve as perfectly fine calibrators, this dampens the claim that the model is doing any kind of meta-reasoning on its own epistemic uncertainty.

I recognize of course that access to such models is limited, and moreover this topic in general is quite thorny and philosophical, so bulletproof answers to these questions need not be established in this paper. But I do think more could have been done to investigate this question, or at least the claims in the paper tempered somewhat (again see Framing).

## Why do this over other calibration methods?
Again, I find the author's use of the term natural language overblown—any existing calibration method can be modified to predict words corresponding to probability ranges, and thus gain the benefits of a "natural language" interface as suggested in paragraph 3 of the introduction.

If not this benefit, then why might one use the verbalized probability method over other existing methods to calibration? The strongest a priori argument is that this method does not require access to model logits, which is true for some models (e.g. retrieval models). I generally buy this claim. But overall, this is a rather niche use case.

I think much more could be done to evaluate the benefits of using q/a semantics and internal representations in place of, or in tandem with, typical features like model logits. For example, Jiang et al., 2021 proposes and evaluates several approaches to pre- and post-hoc calibration of LMs, yet the work is dismissed as "requiring additional overhead" (which is true of the current method too).

---

> ### Author Response · Authors · 2022-07-27
> **Response to "Framing" (in Weaknesses)**
>
> We thank Reviewer oCV4 for the detailed review, useful references, and engagement with our work.
>
>
> **Framing: First paragraph**
>
> It seems Reviewer oCV4 has misunderstood our results (which we realize should be presented in a clearer way). All the results in the main paper are for the setting “Verbalized numbers”, where the model generates tokens for numbers (e.g. “61%”). This is illustrated in Figure 2 (top row). The Verbalized Numbers setting is not 5-class classification, as the model sees many different numbers during fine-tuning (not just 5 numbers).
>
> We also include results for the “Verbalized words” setting in Appendix C.2 and C.3, where the model is fine-tuned to output 5 random names. We don’t include the results for outputting 5 meaningful words (e.g. “low” and “high") instead of random names, but the differences in performance were very small and we are happy to add these results.
>
> Our paper does show something novel. The model learns to output words (either meaningful or random) or percentages via greedy decoding and in each case these are moderately calibrated. So the model is outputting uncertainty in the same way it outputs natural language (in contrast to using the logits). That said, the model is just outputting a single word/number, rather than incorporating uncertainty information into a full sentence or paragraph. So while the model does use natural language, the usage is very constrained. Thus we agree with the reviewer that the sentence is potentially misleading and will consider rephrasing it.
>
>
> **Framing: Second paragraph on use of the word “shown”**
>
> Reviewer oCV4 has taken this sentence out of context. The title of our paper and the first line of the abstract mention “teaching” and “learning” respectively to convey the fact that the model was fine-tuned.
>
> Aside from the wording of our abstract, there is a substantive question about how much the model’s performance when fine-tuned depends on pre-trained latent representations as opposed to features learned during finetuning. S3.4 discusses this question and provides three sources of evidence (regression models, linear probe and projections of GPT3 embeddings, and few-shot results). We do not think our treatment of this is “irresponsible”, as we discuss multiple sources of evidence and say explicitly that the evidence for pre-trained representations is “tentative”.
>
>
>
> **Framing: Third paragraph**
>
> Reviewer oCV4 notes that work by Mielke et al. is “not mentioned”. But this paper came out after our submission. The reviewer also discusses this paper twice in “Requested Changes” (in discussing the novelty of our paper) and includes Mielke et al. as a “Missing Reference”.

---

> > ### Comment · Reviewer_mS55 · 2022-08-06
> > **Agreement with framing concern**
> >
> > Just wanted to note that this reviewer's concerns with the framing overlap with one of my reservations about this paper: I agree that the current framing makes it easy to read the paper as saying "GPT-3 has a metacognitive ability to reason about epistemic uncertainty" or something like that. I think to a large extent this comes from phrasing like models expressing "their uncertainty" or "their own confidence" - this suggests that uncertainty and confidence are things that the model itself has (which in turn suggests that it has things like beliefs and meta-cognition), whereas what we really want is not the model's confidence but rather an expression of how confident users should be in the model.
> >
> > After several re-readings of the intro, I do think careful readers should be able to figure out that the paper is not really arguing for this. But it seems like perhaps the framing should be revised if this misinterpretation is so salient, since readers often can't afford to be careful in the face of the constant flood of ML publications.

---

> > ### Comment · Reviewer_oCZ4 · 2022-08-08
> > **Response**
> >
> > Thanks to the authors for their response to my review! Let me address some points directly, but I'll also try to summarize my updated thoughts and response to the paper in a top-level reply.
> >
> > > It seems Reviewer oCV4 has misunderstood our results (which we realize should be presented in a clearer way). All the results in the main paper are for the setting “Verbalized numbers”, where the model generates tokens for numbers (e.g. “61%”). This is illustrated in Figure 2 (top row). The Verbalized Numbers setting is not 5-class classification, as the model sees many different numbers during fine-tuning (not just 5 numbers).
> >
> > My apologies for not noticing that the main results are for verbalized numbers, but I believe this supports my point, i.e. that I would not call outputting verbalized tokens corresponding to numbers "natural language", insofar as any calibration method could likely be straightforwardly adapted to output wordpiece tokens corresponding to natural numbers, rather than the numbers themselves.
> >
> > > Reviewer oCV4 has taken this sentence out of context. The title of our paper and the first line of the abstract mention “teaching” and “learning” respectively to convey the fact that the model was fine-tuned.
> >
> > I think "teaching" and "learning" are vague terms in the literature, which span the gamut from in-context learning to training from scratch. But I also agree with reviewer Ms55 that the astute reader should quickly notice that the model is finetuned, rather than displaying any "emergent" behavior. I do think the framing of each sentence, especially in the abstract, though, is crucial and must be critically evaluated on its own. This sentence is the "takeaway" sentence which is likely to shape people's framing of the paper as they read it.
> >
> > > Reviewer oCV4 notes that work by Mielke et al. is “not mentioned”. But this paper came out after our submission. The reviewer also discusses this paper twice in “Requested Changes” (in discussing the novelty of our paper) and includes Mielke et al. as a “Missing Reference”.
> >
> > While it is true the peer reviewed version of this work appeared in NAACL 2022, it first appeared on arXiv in Dec 2020. It is not crucial for authors to be aware of every preprint on arXiv, but I would like to see some discussion of the relationship between this work and Mielke et al. 2022, if the paper is accepted.

---

> > > ### Author Response · Authors · 2022-08-13
> > > **Mielke**
> > >
> > > We're sorry for our misunderstanding about the publication date of Mielke. We will add a discussion to the Related Work section.

---

> ### Author Response · Authors · 2022-07-27
> **Response to "Why do [verbalized probability] over other calibration methods?"**
>
> This a response to the last section from "Weaknesses".
>
> Here are some reasons that verbalized probability is potentially valuable:
> 1. Models that use information retrieval may not have logit probabilities over their answers.
> 2. Models can use natural language to express continuous distributions. For example to express uncertainty over questions like “How does wealth vary in this country?” or “When will a large asteroid next hit the Earth?”. With verbalized probability, the model could decide which continuous distribution to use. (If we instead use logits, human users would have to choose the distribution and bounds on the parameters).
> 3. Suppose a non-technical human user Bob wants to get a language model to answer some questions and include uncertainty estimates (e.g. possible strategies for a company and estimates of how likely they are to work). Bob can give some examples of these strategies for few-shot prompting. If the model can produce calibrated verbalized probabilities, then it could usefully generalize the patterns in Bob’s examples. This setup (where Bob can teach the model using natural language) seems more flexible for non-technical users than trying to work with the logits.
> 4. It’s possible that verbalized probability calibration will sometimes generalize better than logit-based calibration. We showed one example (generalizing to the Multi-Answer evaluation set).
>
> We do not discuss 1-4 in much detail in the paper. Our goal is to introduce verbalized probability and show a concrete case-study. We don’t think verbalized probability needs to be superior to logit-based confidence methods in order to be worth the ML community thinking about. (Indeed, we think logit-based methods are also extremely valuable and we expect there are usefful ways to combine verbalized probability and logits.)

---

> ### Author Response · Authors · 2022-07-27
> **Has GPT-3 learned to express its own uncertainty? Is this a "very bold" claim?**
>
> It seems like a key issue for Reviewer oCV4 is whether GPT-3 has actually learned to express its *own* uncertainty using words/numbers. This issue is raised both in “Framing” and in the “Are you sure GPT-3 isn’t using superficial statistics…” sections.
>
> First we agree that we do not provide definitive evidence that GPT-3 has learned to express its own uncertainty and that much further work needs to be done. (We note in S3.4 that evidence is “tentative” and S4.1 includes suggestions for future work).
>
> We will give an overview of our position.
>
> The claim that “GPT-3 can learn to express its own uncertainty in words/numbers” is not actually “very bold”. Contrary to what Reviewer oCV4 says, it doesn’t imply a general ability for “metacognition” or “meta-awareness”.
>
> Our results show that GPT-3’s logits are moderately calibrated zero-shot on both the Add-subtract and Multiply-divide task. The logits contain information about GPT3’s *own* uncertainty about the next token, not human uncertainty or the uncertainty of another model. During our finetuning for verbalized probability, GPT3 could learn to output verbalized probabilities for an answer that depend on its zero-shot logit for the answer. (To be more precise, both the logit and the verbalized probability could depend on the same feature or pre-trained representations.) And if the verbalized probability depends on the logits, then it expresses the model’s own uncertainty. This is a fairly simple mechanism and doesn’t imply that GPT-3 would be capable of more sophisticated kinds of metacognition and self-knowledge.
>
> Of course, we argued in S4.3 that GPT-3 does not simply learn to output its logits (for verbalized probability), as this would lead to worse performance on the Multi-answer evaluation set than we observe. However, it’s still plausible that GPT-3’s verbalized probabilities are a function of its logits. In addition, GPT-3 could learn to use the following information when giving a confidence for a question-answer (Q,A):
>
> 1.  GPT-3’s logit distribution over alternative answers to Q. For instance, if Q=”Name an even number below 10” the logits might be a uniform distribution on {2,4,6,8} and ~0 elsewhere.
> 2.  GPT-3’s logit distribution over the tokens in the question Q, which would measure how much Q is unpredictable or OOD for GPT-3.
> 3.  Features of the question Q itself. For example, does Q include large or small integers?
>
> Examples 1 and 2 in this list are internal features of GPT-3 that reflect its own uncertainty (like the logit on answer A). So if GPT-3 learns to use them, it will be expressing its own uncertainty. By contrast, if GPT-3 used only features like 3, it would not be expressing its own uncertainty.
>
> Finally, it’s possible that GPT-3 does have latent “emergent” representations that relate to epistemic uncertainty (i.e. subjective uncertainty about whether a claim is true or false) and that are not closely related to the logits. We know that small RNN models learn representations of sentiment (“sentiment neurons”) as this latent feature is useful for predicting the next word. For GPT-3, it would be useful to represent the likelihood of claims being true or false, as this would help predict the next word in contexts where truth is important (e.g. science textbooks, Wikipedia). [We focus on the logits in the preceding paragraphs, because it’s a concrete mechanism we know relates to the models own uncertainty.]
>
> Again, we do not provide strong evidence that GPT-3’s verbalized probabilities are a function of its zero-shot logits on either questions or answers (or other features relating to epistemic uncertainty). However, we think it’s plausible a priori and we do provide some suggestive evidence in S3.4, namely:
>
> 1. 50-shot performance is fairly close to fine-tuning on 10k examples. This suggests that GPT-3 is mainly using existing features to produce verbalized probability, because it cannot learn much from 50 examples. (NB: There are ~100 categories of question in training, and so 50 examples is very limited).
> 2. Evidence from the generalization performance of the linear probe and two-dimensional projection on top of GPT-3 embeddings. (The embeddings contain features related to the zero-shot logit).

---

> > ### Author Response · Authors · 2022-08-13
> > **This comment was not addressed in detail by the reviewer**
> >
> > This comment highlighted results in our paper and some additional considerations to respond to (what we felt) was the reviewer's main objection. The reviewer does not really address this comment in their followup. (We realize our comment was long and involved).

---

> ### Comment · Reviewer_oCZ4 · 2022-08-08
> **Overall response**
>
> Thanks to the authors for their detailed response to my review which has clarified my view on the paper.
>
> Overall, what I'm trying to express with the somewhat vague concerns above is that I feel the paper could have more thoroughly explained and investigated how verbalized probability is a different *class* of calibration method. I got the sense that the authors were pushing this view with the framing (e.g. "This is the *first time* a model has been shown to express uncertainty in natural language"), and given how they did/did not contextualize their work with the prior calibration literature.
>
> Insofar as I view the probability -> "natural language" mapping in this paper trivial, I do *not* believe this is the "first time" a model has been shown to express uncertainty in "natural language" as defined by this paper; as I said, any calibration method might fit the bill here. Yet I believe it would be weird for the authors of the original Platt scaling paper to say that "this is the first time a model has been shown to express its own uncertainty (in numbers)." Calibration is a post-hoc transformation you do on a model to better measure *user* confidence in a model (I am in agreement with reviewer ms55 on this). The language here (“shown”, “express”, “own”) seems to imply otherwise.
>
> The key novelty in this work, as the authors write, is that the model learns to output calibration estimates the same way it outputs, well, everything–i.e. through the language decoding channel of an LM. I **do** agree this is a potentially fascinating and important idea but I think the paper could have done more to discern whether this is actually an important differentiator over the existing work on calibration—to show that this is not just a calibration method with “extra steps”. Two of the contributions that this paper stands to make are:
>
>  1. Does verbalized probability enable new model interactions and applications that traditional calibration doesn’t support? I agree with the reviewers’ comments on the potential benefits of using a natural language interface for uncertainty, and I agree that verbalized probability does not need to be superior to existing calibration methods if there auxiliary (user-facing) benefits. Yet the paper unfortunately uses a very minimal/sparse definition of natural language (either percentages expressed in tokens, or 5-class bins) and doesn’t get at the more interesting uses of language, e.g. to express more unbounded/vague certain categorizations. I do think the strongest contribution in this regard is the few-shot learning experiments, which show that you can get verbalized probability estimates without finetuning.
>   2. How important is it actually for the model to output **its own** calibration estimates  **in the same way it outputs language**? If I view this paper as a calibration paper, it is essentially proposing a technique for finetuning a model to produce calibration estimates given pretrained semantic information. These could be estimates of itself, or indeed any other model. Whether it’s important for the model to be “calibrating itself” is a key unanswered question that I believe is a prerequisite for the paper’s framing of “a model estimating its own uncertainty”, etc. This is precisely why I suggested the other baselines (an equal-capacity but non-trained LM, a different pretrained LM, etc) to more clearly understand whether there are certain “meta-cognitive” features in GPT-3 that indeed are being used for uncertainty estimation.
>   The interpretability analyses (page 9) get at this question by attempting to show that the representations encode uncertainty information (via probing), but I would also be interested in to what extent the probing performance can be attained from any pretrained model trying to estimate GPT-3’s accuracy. Moreover, the more direct evaluation would be to just literally use a different/untrained LM for calibration and show that it does (or does not) estimate probabilities as well.
>
> To summarize, I believe as it stands the paper’s framing should be much more similar to a standard calibration paper, especially [Zhang et al. 2021](https://arxiv.org/abs/2106.01494), in proposing an approach to calibrating LMs that makes use of pretrained semantics. I think the paper has valuable experiments and interesting results that elevate the paper beyond the existing work, but I think it’s important to situate the work more squarely in this literature.
>
> If the paper keeps its current framing, which the authors seem inclined to do (but feel free to clarify), I think it’s much more important to more convincingly show (1) it’s important to have a model "express its own uncertainty" and (2) the use of “natural language” here enables more exciting applications than simply attaching a “number -> tokenization” head onto any existing calibration method. As it stands, to get my unequivocal recommendation, I would want some follow up experiments on either points 1 or 2 above.

---

> > ### Author Response · Authors · 2022-08-13
> > **Main issues: (1) the language we used in the paper, (2) our experiments not being extensive enough (though we did many)**
> >
> > A major concern for the reviewer is the framing of the paper and the specific language we used. We recognize there is room for disagreement about how the paper is framed. Yet we think our paper is clear about what tests are being done and the limitations of our results (including in the abstract and introduction). Thus, readers would be able to judge for themselves whether or not they disagree with our framing and language.
> >
> > The reviewer says, “the paper could have done more to discern whether [expressing uncertainty in words]” is an important differentiator” and suggests a number of ideas for additional experiments. These are good suggestions for future research. But in asking for these additional experiments, we think the reviewer is choosing a very high bar for a research contribution.
> >
> > Our paper includes a new task (CalibratedMath) which took time and multiple iterations to design. Using this task, we carried out the following experiments:
> >
> > 1. Finetuning GPT-3 (175B parameter model)
> > - We tried two different training sets (Add-subtract, Multiply-divide).
> > - We did both verbalized probability and indirect logits.
> > - For verbalized probability, we tried numbers, words, and random names.
> >
> > 2. Finetuning heuristic models
> > - Logistic regression on heuristic features
> > - Linear probe on GPT-3 embeddings
> > - 2D projection on embeddings
> > - MLP on the embeddings
> >
> > 3. Few-shot (up to 50 shot)
> >
> > Evaluation
> > - All models were evaluated on two different eval sets.
> >  - We also evaluated the zero-shot logit on these two eval sets.
> > - We evaluated both with EV and greedy decoding.
> > - We evaluated finetuned models at different checkpoints.
> >
> > We are limited in how many experiments we can run and our ability to train/finetune large LMs. We made a choice to do a thorough investigation of a particular task (namely simple math questions). We think that a thorough investigation of one task is easier for other researchers to build on than a more superficial coverage of many tasks. The reviewer's points (1) and (2) above raise important questions about verbalized probability in LMs. Our paper discusses both of these points (especially 2) and provides  experiments to shed (some) light on them. Because of limits on how many experiments we can run and on space in the paper, we cannot address them in the level of detail the reviewer would like. But we think our paper suffices for a first paper on this topic.

---

### Decision · Action_Editors · 2022-09-04

**Recommendation:** Accept with minor revision

**Comment:**

The work is clearly of interest to the community.  The question of correctness can be interpreted in two ways: 1. Accuracy of presented results, 2. how accurately the paper's claims are supported by the evidence presented.  The former is true, while the latter is slightly debated by the reviewers.   A number of concrete suggestions have been presented to improve the paper.

The required modification is that the paper's framing be updated to clearly align the results as an extension of existing calibration work and as multiple reviewers read the paper as implying "metacognitive" abilities, the introduction should address this directly.  There are a number of places that likely move the reader to make this inference, e.g. use of terms like "human-like", which should be avoided. See related comments around “shown”, “express”, and “own”.

A number of other requests (e.g. Oracles for better understanding of the results) would also improve the work.

---

> ### Author Response · Authors · 2022-09-15
> **Thanks**
>
> We'd like to thank the AE and all reviewers for feedback, discussion, and suggestions. We will make these revisions.

---

> > ### Author Response · Authors · 2022-10-03
> > **Summary of revisions in updated version**
> >
> > We have submitted a camera-ready version with the following revisions.
> >
> > Changes to the Introduction:
> > We removed the description of verbalized uncertainty as “human-like”, as this was misleading and potentially suggested “meta-cognitive” abilities. We adjusted the relevant paragraph to clearer. We also clarify that our contribution is only taking a "first step" towards expressing uncertainty in natural language (because verbal probabilities are expressed with a single token).
> >
> > We added a new paragraph clarifying the contribution of the paper. We note that our results are closely related to work that finetunes the model logits to be better calibrated (and we cite this work).
> >
> > Changes to abstract
> > We edited the abstract to make the framing more accurate and less likely to be misleading (e.g. to reduce the chance that readers think we are making claims about meta-cognitive abilities of models).
> > We removed a sentence from the Abstract about novelty of the work. We also added discussion of how our paper relates to Mielke et al. in the Related Work section.
> >
> > Change to Results:
> > Remove a claim from Section 3.4 that was potentially misleading and suggested “meta-cognitive” abilities on the part of finetuned models.
> >
> > Change to Discussion:
> > We add a discussion of the “cross-model” calibration experiments that were discussed in the reviews. We attempted such experiments but they were confounded because alternative models available to us had much lower capacity than GPT-3. We add a reference to a subsequent paper that ran this kind of experiment.
> >
> > Related Work:
> > We add a discussion of Mielke et al (2022) and how it relates to our paper.
> >
> > Acknowledgements (camera ready). We will add an acknolwedgment to the reviewers for very helpful feedback.